# Score and Lyrics-Free Singing Voice Generation

## Abstract

Generative models for singing voice have been mostly concerned with the task of "singing voice synthesis," i.e., to produce singing voice waveforms given musical scores and text lyrics. In this work, we explore a novel yet challenging alternative: singing voice generation without pre-assigned scores and lyrics, in both training and inference time. In particular, we propose three either unconditioned or weakly conditioned singing voice generation schemes. We outline the associated challenges and propose a pipeline to tackle these new tasks. This involves the development of source separation and transcription models for data preparation, adversarial networks for audio generation, and customized metrics for evaluation.

## 1 Introduction

The task of computationally producing singing voices is usually referred to as singing voice *synthesis* (SVS) in the literature (Cook, 1996). Most researchers assume that the note sequence and the lyrics of the waveform to be generated are given as the model input, and aim to build synthesis engines that sound as natural and expressive as a real singer (Blaauw et al., 2019; Hono et al., 2019; Kaewtip et al., 2019; Lee et al., 2019a; Tamaru et al., 2019). As such, the content of the produced singing voice is largely determined by the given model input, which is usually assigned by human. And, accordingly, progress in SVS has followed closely with that in text-to-speech (TTS) synthesis (Umbert et al., 2015; Shen et al., 2017; Gibiansky et al., 2017).

However, we argue that singing according to a pre-assigned musical score and lyrics is only a part of the human singing activities. For human beings, singing can also be a spontaneous activity. We learn to spontaneously sing when we were children (Dowling, 1984). We do not need a score to sing when we are humming on the road or in the bathroom. The voices sung do not have to be intelligible. Jazz vocalists can improvise according to a chord progression, an accompaniment, or even nothing.

We aim to explore such a new task in this paper: teaching a machine to sing with a training collection of singing voices, but without the corresponding musical scores and lyrics of the training data. Moreover, the machine has to sing without pre-assigned score and lyrics as well even in the inference (generation) time. This task is challenging in that, as the machine sees no lyrics at all, it hardly has any knowledge of the human language to pronounce or articulate either voiced or unvoiced sounds. And, as the machine sees no musical scores at all, it has to find its own way learning the language of music in creating plausible vocal melodies. It also makes the task different from TTS.

Specifically, we consider three types of such score- and lyrics-free singing voice *generation* tasks, as shown in Figures 1(b)–(d). A *free singer* sings with only random noises as the input. An *accompanied singer* learns to sing over a piece of instrumental music, which is given as an audio waveform (again without score information). Finally, a *solo singer* also sings with only noises as the input, but it uses the noises to firstly generate some kind of 'inner ideas' of what to sing.

From a technical point of view, we can consider SVS as a *strongly conditioned* task for generating singing voices, as the target output is well specified by the input. In contrast, the proposed tasks are either *unconditioned* or *weakly conditioned*. This work therefore contributes to expanding the "spectrum" (in terms of the strength of conditional signals) of singing voice generation. Doing so has at least two implications. First, while our models are more difficult to train than SVS models, they enjoy more freedom in the generation output. Such freedom may be desirable considering the artistic nature of singing. Second, we can more easily use a larger training set to train our model—

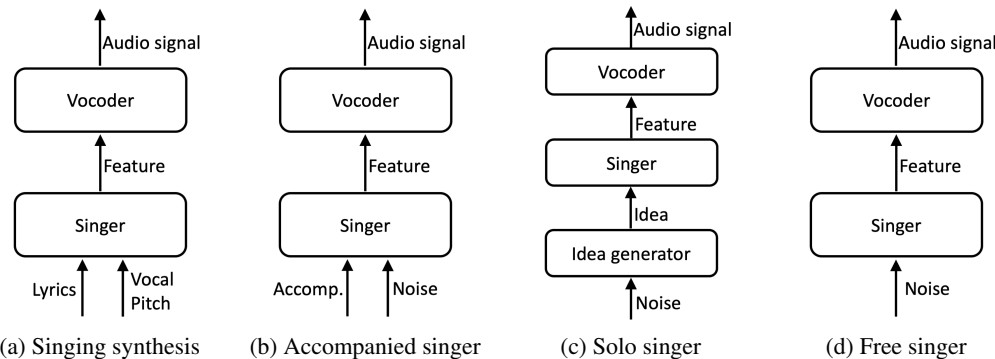

Figure 1: Schemes of singing voice generation; the strength of condition decreases from left to right.

due to the difficulty in preparing time-aligned scores and lyrics, the training set employed in existing work on SVS usually consists of tens of songs only (Lee et al., 2019a); in contrast, in our case we do not need labeled and aligned data and can therefore use more than hundreds of songs for training. This may help establish a universal model based on which extensions can be made.

The proposed accompanied singer also represents one of the first attempts to produce singing voice given an accompaniment. One intuitive approach to achieve this is to first generate a score according to an accompaniment in the symbolic domain and then synthesize the singing voices according to the score. The second step of synthesis is relatively well-established, but the first step of generating a score given an accompaniment is not explored yet. Extensive researches have been done in generating scores of one or several instruments (Hadjeres et al., 2017; Yang et al., 2017; Huang et al., 2019; Payne, 2019). However, to the best of our knowledge, very few, if any, researches have been done on generating scores of singing voices given an accompaniment. Our approach bypasses the step of generating scores by directly generating the mel-spectrogram representation.

We outline below the challenges associated with the proposed tasks and the solutions we investigate.

First, the tasks are unsupervised as we do not provide any labels (e.g., annotations of phonemes, pitches, or onset times) for the training singing files. The machine has to learn the complex structure of music directly from audio signals. We explore the use of generative adversarial network (GAN) (Goodfellow et al., 2014) to address this issue, for its demonstrated effectiveness for SVS (Hono et al., 2019) and pitch-conditioned instrument note synthesis (Engel et al., 2019). Specifically, we design a novel GAN-based architecture to learn to generate the mel-spectrogram of singing voice, and then use WaveRNN (Kalchbrenner et al., 2018), a single-layer recurrent neural network, as the vocoder to generate the audio waveform. Rather than considering the mel-spectrograms as a fixed-size image as done in recent work on audio generation (Engel et al., 2019; Marafioti et al., 2019), we use gated recurrent units (GRUs) (Cho et al., 2014) and dilated convolutions (van den Oord et al., 2016) in both the generator and discriminator, to model both the local and sequential patterns in music and to facilitate the generation of variable-length waveforms.

Second, for training the free singer, unaccompanied vocal tracks are needed. As for the accompanied singer, we need additionally an accompaniment track for each vocal track. However, public-domain multi-track music data is hard to find. We choose to implement a vocal source separation model with state-of-the-art separation quality (Liu & Yang, 2019) for data preparation. The proposed pipeline for training and evaluating an accompanied singer is illustrated in Figure 2. The advantage of having a vocal separation model is that we can use as many audio files as we have to compile the training data. The downside is that the singing voice generation models may suffer from the artifacts (Cano et al., 2018) of the source separation model, which is moderate but not negligible.[1]

Third, for the accompanied singer, there is no single "ground truth" and the relationship between the model input and output may be one-to-many. This is because there are plenty of valid ways to

---

[1]Research on singing voice separation has made remarkable progress in recent years. As reported in (Stöter et al., 2018), state-of-the-art models perform comparably to an oracle approach for many cases. Open-source implementations of state-of-the-art models are also available (Stöter et al., 2019; Hennequin et al., 2019). Visit https://bit.ly/2Xattua for samples of the separation result of the model we employ.

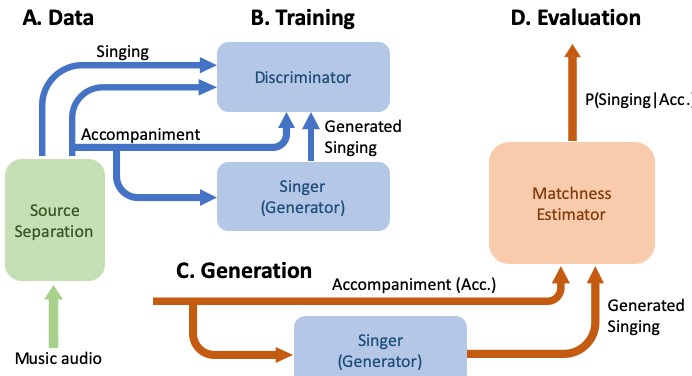

Figure 2: A pipeline for building the accompanied singer. We use source separation to get separated singing voice and accompaniment from professionally recorded audio files. Then, we use the separated tracks to train the generators and discriminators in the GAN. In inference time, we feed an unseen accompaniment to the trained singer model and let it "sing."

sing over an accompaniment track. For diversity and artistic freedom, we cannot ask the machine to generate any specific singing voice in response to an accompaniment track, even if we have paired data of vocal and accompaniment tracks. We investigate using conditional GAN (Mirza & Osindero, 2014) to retain the possibility of generating singing voices with multiple modes.

Fourth, as the proposed tasks are new, there are no established ways for performance evaluation. According to our setting, we desire our machine to generate audio waveforms with high quality and diversity, vocal-like timbre, plausible pitch contour, emotion expression, and, for the accompanied singer, that are in harmony with the given accompaniment track. But, the singing does not have to be intelligible. We propose customized objective and subjective metrics to evaluate our models in these aspects. For example, we adapt the melody harmonization model proposed by Lim et al. (2017) to measure the matchness between the generated vocal track and the given accompaniment track.

Finally, reproducibility is a major issue, especially for a subjective task. We intend to use publicly-available copyright-free instrumental music as the conditional signals for testing the accompanied singer, so that other researchers can use the same testing conditions for model comparison in the future. We will also release the testing conditions for the solo singer, the generated singing voices for all our models, as well as open source our code through a public git repository [URL removed].

We focus on Jazz music in this work. Samples of the generated singing voices can be found at https://bit.ly/2mIvoIc. Our models have many possible use cases. For example, we can use the accompanied singer as a backing vocalist. In addition, we can use the free singer as a sound source—to demonstrate this, we make a song by hand in the style of Jazz Hiphop by sampling the output of our free singer. This song can be listened to at https://bit.ly/2QkUJoJ.

## 2 SCHEMES OF SINGING VOICE GENERATION

A **free singer** takes no conditions at all as the input. We want it to sing freely. The singing voices from a free singer may not even sound good, but they should sound like singing voice. A free singer is like we are freely humming or singing on the road walking or in the bathroom taking a shower. We may not even know what we are singing and likely there is no underlying musical score.

From the viewpoint of a generative model, training a free singer amounts to modeling a distribution $P(\mathbf{Y})$, where $\mathbf{Y} \in \mathbb{R}^{K \times T}$ is a matrix representing a sequence of $K$-dimensional features and $T$ is the number of time frames. A free singing is sampled from this distribution without conditions.

An **accompanied singer** takes as the input a sequence of accompaniment-derived features. An accompanied singer tries to generate singing voices that match the accompaniment track in some way. It is similar to the case of Karaoke, where a backing accompaniment track is played from a speaker, the lyrics and a video are displayed on a screen, and a user tries to sing according to the

lyrics and the backing track. The difference is that, this time the user is a trained model and we do not ask it to follow the lyrics or the exact pitch contour of the accompaniment. The note sequence found in the singing has to be in harmony with, but not a duplicate of, that in the backing track.

Training an accompanied singer amounts to modeling a distribution $P(\mathbf{Y}|\mathbf{A})$, where $\mathbf{Y} \in \mathbb{R}^{K \times T}$ represents a feature sequence of the vocal track, and $\mathbf{A} \in \mathbb{R}^{H \times T}$ is a feature sequence of the given accompaniment track. In our implementation, we use the mel-spectrograms for $\mathbf{Y}$ in compliance with the need of the vocoder. For $\mathbf{A}$, different features can be tried, and we investigate using a transcription model (Hawthorne et al., 2018) to extract pitch features. See Section 4.1 for details.

A **solo singer** is similar to a free singer in that both takes no conditions as the input. However, a solo singer would generate an 'inner idea' first, and then sing according to that. In other words, it learns a joint distribution $P(\mathbf{Y}|\mathbf{I})Q(\mathbf{I})$, where $\mathbf{I} \in \mathbb{R}^{J \times T}$ is a matrix representing the idea sequence. The singer first samples $\mathbf{I}$ from a distribution $Q$, and then uses that to conditionally sample $\mathbf{Y}$ from $P$. The inner idea $\mathbf{I}$ can take several forms. In this work, we instantiate this scheme with $\mathbf{I}$ being a chord progression (namely a sequence of chord labels). The distribution $Q$ is modeled by an auto-regressive recurrent network we build for chord progression generation (with a network architecture adapted from that in (Waite et al., 2016)), as described in Section 3.3.

Alternatively, we can think of a solo singer as a combination of an idea generator and an an accompanied singer. For an accompanied singer, the information extracted from the given accompaniment track can take several forms such as transcribed pitches and chord progressions. A solo singer learns to generate such information on its own, without reference to an actual accompaniment track.

## 3 MODELS

To account for the absence of supervised data and the highly complicated spatio-temporal patterns in audio spectrograms, we propose a new adversarial net that features heavy use of GRUs (Cho et al., 2014; Chung et al., 2014), dilated convolutions (van den Oord et al., 2016; Sercu & Goel, 2016), and feature grouping to build our singer models. We provide the algorithmic details below.

### 3.1 BLOCK OF GRU-GROUPED DILATED CONVOLUTION-GROUP NORMALIZATION

Network architectures with stacked blocks of GRUs and dilated convolutions have been used to attain state-of-the-art performance in blind musical source separation (Liu & Yang, 2019). In a source separation task, a model learns to decompose, or unmix, different sources (e.g., vocal, piano, bass, drum) from a mixture signal (Rafii et al., 2018). This requires the abilities to model the relationships between different sources as well as the relationships between neighboring time frames. The output spectrograms are also expected to be distortion-less and of high audio quality. For it has demonstrated its capability in source separation, we adopt it as a building block of the singer models. Especially, we want the singer models to also consider accompaniment information.

Specifically, one such block we adopted in our models is a stack of GRU, dilated convolution with feature grouping, and group normalization (Wu & He, 2018). The input to the GRU, the output of the GRU, and the output of the group normalization are summed to form the output of the block. We note that the original 'D2 block' used in (Liu & Yang, 2019) uses dilated GRU and uses weight normalization (Salimans & Kingma, 2016) for the dilated convolution layers. However, empirically we find that it is easier for the singer models to converge by replacing weight normalization with group normalization, and using plain GRUs is as good as using dilated GRUs. We refer to our blocks as GRU-grouped dilated convolution-group normalization block ('G3 block').

### 3.2 SINGER MODELS WITH BEGAN, G3 BLOCKS AND FRAME-WISE NOISES (G3BEGAN)

The accompanied singers and solo singers have to take conditions as part of their input. One desirable property of the models is the ability to generate voices with arbitrary length, as the conditional signal can be of variable length. Besides, the model has to deal with the one-to-many issue mentioned in Section 1, and the absence of supervisory signals. With these issues in mind, we design a GAN architecture for score and lyrics-free voice generation. In particular, we pay special attention to the following three components: 1) the network architecture, 2) the input noises for GAN, and 3) the loss function of the discriminator.

Let us first take a look at two existing GAN models for audio generation: (Engel et al., 2019) and (Donahue et al., 2019). Their generators and discriminators are both based on 2D convolutions, transposed 2D convolutions and dense (linear) layers. The generators take a vector $\mathbf{z} \in \mathbb{R}^U$ as the input noise and use transposed convolutions to expand $\mathbf{z}$ so that a temporal dimension emerges in the expanded intermediate matrices. The number of temporal frames in the final output depends on the total strides used in all the transposed convolutions. The discriminators take the output of the generators or the real signal as the input, and compress the input matrix with convolution layers until the output becomes a single value represents the prediction of true (real) or false (generated) data.

A main reason why existing models cannot generate variable-length output is the need to expand $\mathbf{z}$ by transposed convolution layers. We remedy this by using an architecture consisting of the proposed G3 blocks, and convolutions without strides, for both the generators $G(\cdot)$ and discriminators $D(\cdot)$. Moreover, instead of using a single noise vector, our models take as input a sequence of noise vectors, denoted as $\mathbf{Z} \in \mathbb{R}^{U \times T}$, that has the same temporal length as the desired output $\mathbf{Y}$. Each column of $\mathbf{Z}$ is sampled independently from a Gaussian distribution $Normal(0, 1)$. At the first glance, it might feel unnatural to have one noise vector per frame as that may result in fast oscillations in the noises. However, we note that the output of $G(\cdot)$ for the $t$-th frame depends not only on the $t$-th column of $\mathbf{Z}$ (and $\mathbf{C}$ or $\mathbf{I}$), but the entire $\mathbf{Z}$ (and the condition matrices), due to the recurrent GRUs in the model. We expect that the GRUs in the discriminator $D(\cdot)$ would force $G(\cdot)$ to generate consistent consecutive frames. Therefore, the effect of the frame-wise noises might be introducing variations to the generation result (e.g., by adjusting the modes of the generated frame-wise features).

As for the loss function of $D(\cdot)$, we experiment with the following three options: the vanilla GAN, the LSGAN (Mao et al., 2017) that adopts the least squares loss function for the discriminator, and the boundary equilibrium GAN (BEGAN) (Berthelot et al., 2017) that adopts an "auto-encoder style" discriminator loss. The $D(\cdot)$ in either GAN or LSGAN is implemented as a classifier aiming to distinguish between real and generated samples, whereas the $D(\cdot)$ in BEGAN is an autoencoder aiming to reconstruct its input. Specifically, in BEGAN, the loss functions $l_D$ and $l_G$ for the discriminator and generator, as in the case of the accompanied singer, are respectively:

$$l_D = L(\mathbf{X}, \mathbf{C}) - \tau_s L(G(\mathbf{Z}, \mathbf{C}), \mathbf{C}), \tag{1}$$

$$l_G = L(G(\mathbf{Z}, \mathbf{C}), \mathbf{C}), \tag{2}$$

where $\mathbf{X} \in \mathbb{R}^{K \times T}$ is the feature sequence of a real vocal track sampled from the training data, $G(\mathbf{Z}, \mathbf{C}) \in \mathbb{R}^{K \times T}$ is the feature sequence for the generated vocal track, and $L(\cdot)$ is a function that measures how well the discriminator $D(\cdot)$, implemented as an auto-encoder, reconstructs its input:

$$L(\mathbf{M}, \mathbf{C}) = \frac{1}{WT} \sum_{w,t} |D(\mathbf{M}, \mathbf{C})_{w,t} - M_{w,t}|, \quad \text{for an arbitrary } W \times T \text{ matrix } \mathbf{M}, \tag{3}$$

where we use $M_{w,t}$ to denote the $(w,t)$-th element of a matrix $\mathbf{M}$ (and similarly for $D(\mathbf{M}, \mathbf{C})_{w,t}$). Moreover, the variable $\tau_s$ in Eq. (1) is introduced by BEGAN to balance the power of $D(\cdot)$ and $G(\cdot)$ during the learning process. It is dynamically set to be $\tau_{s+1} = \tau_s + \lambda(\gamma L(\mathbf{X}, \mathbf{C}) - L(G(\mathbf{Z}, \mathbf{C}), \mathbf{C}))$, for each training step $s$, with $\tau_s \in [0, 1]$. $\lambda$ and $\gamma$ are manually-set hyperparameters.[2]

Empirical comparison of the performance of GAN, LSGAN and BEGAN can be found in Appendix D. It turns out that the BEGAN-based one, referred to as G3BEGAN hereafter, works the best.

### 3.3 CHORD PROGRESSION GENERATOR

We also use auto-regressive RNN to build a chord progression generator for implementing the solo singer. Our chord generator is trained on the Wikifonia dataset (`http://www.wikifonia.org/`), a set of 6,670 songs in the leadsheet format (i.e., with separated symbolic melody and chord tracks). Its chord vocabulary covers 612 different chords. We set the harmonic rhythm of the chord generator such that a chord change may occur every beat. We desire the chord generator to freely generate chord progressions across different tempo values, time signatures, and keys. Moreover, the generated chord progression has to be rhythmically correct. In achieving so, we encode the tempo,

---

[2]We replace $\mathbf{C}$ in the above equations by an empty matrix for the free singer, and by $\mathbf{I}$ for the solo singer. For the accompanied singer, we note that $D(\cdot)$ has two ways to discriminate true and fake data: from whether $G(\mathbf{Z}, \mathbf{C})$ sounds like real singing voice, and from whether $G(\mathbf{Z}, \mathbf{C})$ matches the given condition $\mathbf{C}$.

time signatures, and key information (which are available in the Wikifonia dataset) as the initial hidden state of the RNN, and concatenate the chord vector from last time step with the beat position (e.g., 1, 2, 3, 4) of that time step as the input to the RNN. For data augmentation, we transpose the chord progressions found in Wikifonia to 12 possible keys. Once trained, we use the chord generator to create chord progressions for testing the solo singer. With a one-hot encoding, each column of the condition matrix $\mathbf{I}$ would have dimension $J = 660$.

More details of G3BEGAN and the chord generator can be found in Appendices A.1 and A.2.

## 4 EXPERIMENTS

### 4.1 EXPERIMENTAL SETUP AND DATASETS

In our implementation, we use 80-dimensional mel-spectrograms as the acoustic features modeled and generated by the singer models (i.e., $K = 80$). We use the python package `librosa` (McFee et al., 2015), with default settings, to compute the mel-spectrograms from audio. A mel-spectrogram is passed to a WaveRNN vocoder (Kalchbrenner et al., 2018) to generate an audio signal from mel-spectrograms. Our implementation of the WaveRNN vocoder is based on the code from Fatchord.[3] Instead of using off-the-shelf pre-trained vocoders, which are typically trained for TTS, we train our vocoder from scratch with a set of 3,500 vocal tracks we separate (by a separation model) from an in-house collection of music that covers diverse musical genres.

One main difficulty in conducting this research is to get vocal tracks for training our singer models. Existing multitrack datasets that contain clean vocal tracks are usually diverse in musical genre and the singing timbre, making it hard for our models to learn the relationship between singing and accompaniment. As we set out to focus on Jazz music in this work, we opt for collecting our own vocal tracks from Jazz music only. We therefore implement our source separation model following the architecture proposed by Liu & Yang (2019), which represents the state-of-the-art as evaluated on the MUSDB benchmark (Rafii et al., 2017). We use the whole MUSDB dataset to train our separation model. This dataset contains clean vocal and accompaniment tracks. However, for constructing the condition matrix $\mathbf{A}$ of our accompanied singer, we desire to have separated piano tracks as well (we will explain why shortly). We therefore collect additionally 4.5 hours of Jazz piano solo audio and use them to augment MUSDB for training our source separation model, which can as a result isolate out not only the vocal track but also the piano track from an arbitrary song.

We collect 17.4 hours of Jazz songs containing female voices and 7.6 hours of Jazz songs with male voices. We use the aforementioned separation model to get the vocal tracks. For batched training, we divide the tracks into 10-second sub-clips. Sub-clips that contain less than 40% vocals, as measured from energy, are removed. This leads to 9.9-hour and 5.0-hour training data for female and male Jazz vocals, respectively. 200 and 100 sub-clips are reserved from the training set as the validation set for female singing and male singing, respectively. Each model is trained for 500 epochs. For GAN and LSGAN, we use the models at the 500th epoch for evaluation. For BEGAN, the parameters of the epoch with the best convergence rate (Berthelot et al., 2017) are used for evaluation.

For the accompanied singer, we experiment with extracting pitch-related information from the accompaniment track to form the matrix $\mathbf{A}$ that conditions the generation of the vocal track. The assumption here is that whether the generated vocal track is in harmony with the accompaniment track can be largely determined by pitch-related information. For this purpose, we implement a piano transcription model to transcribe the separated piano track, leading to 88-dimensional transcribed frame-wise pitch as the accompaniment condition (i.e., $H = 88$, as there are 88 piano notes). We implement a piano transcription model with the G3 blocks introduced in Section 3.1, following the training procedure of (Hawthorne et al., 2018). We also implement the model of Hawthorne et al. (2018). Under the same training setting, we find that ours is slightly worse than theirs in note F1 score (0.779 vs 0.794) but slightly better in the note precision score (0.834 vs 0.823). We decide to use our model for the better precision, but the difference of using either of them should be small.

The clips in the training set of our singer models may not contain piano playing (see Table 5 for a summary of the datasets). Even if a clip contains piano playing, the piano may not play across the entire clip. Hence, the models have to learn to sing either with or without the piano accompaniment.

---

[3] https://github.com/fatchord/WaveRNN

For performance evaluation, we collect 5.3 hours of Jazz music from Jamendo (`https://www.jamendo.com`), an online platform for sharing copyright-free music. As said in Section 1, this test set is meant to be public. We apply source separation to the audios, divide each track into 20-second sub-clips,[4] and remove those that do not contain piano. Piano transcription is also applied to the separated piano track, yielding 402 20-second sub-clips for evaluation. 402 chord progressions are generated by the chord generator to evaluate the solo singer and to compute the matchness.

## 4.2 BASELINES

As this is a new task, there is no previous work that we can compare with. Therefore, we establish the baselines by 1) computing the baseline objective metrics (see Section 4.3) from the training data of the singing models, and 2) using existing SVS systems for synthesizing singing voices.

For the SVS baselines, we employ Sinsy (Oura et al., 2010; Hono et al., 2018) and Synthesizer V (Hua et al., 2019), the two well-known SVS systems that are publicly accessible. For Sinsy, we use the publicly available repository[5] to query the Sinsy API (`http://sinsy.jp/`); we use the HMM version (Oura et al., 2010) instead of the deep learning version as the latter cannot generate male voices. For Synthesizer V, we use their software (`https://synthesizerv.com/`). We use Sinsy for both objective and subjective tests, but Synthesizer V for subjective test only, for the latter does not provide a functionality to batch process a collection of MIDI files and lyrics.

SVS systems have to take lyrics and a melody to synthesize singing voices. For the lyrics, we choose to use multiple 'la,' the default lyrics for Synthesizer V.[6] For the melodies, we consider two methods:

1. Vocal transcription from singer training data. We use CREPE to transcribe the separated vocals from the singer training data, and convert it to MIDI format.

2. Piano transcription from the Jamendo testing data. As described in Section 4.1, we have separated and transcribed the piano part of the Jamendo data. Yet, the piano transcription often contains multiple notes at the same time. We use the skyline algorithm (Ozcan et al., 2005) to the transcription result to get a melody line comprising the highest notes.

## 4.3 OBJECTIVE METRICS AND OBJECTIVE EVALUATION RESULT

The best way to evaluate the performance of the singer models is perhaps by listening to the generated results. Therefore, we encourage our readers to listen to the audio files provided in the supplementary material. However, objective evaluation remains desirable, either for model development or for gaining insights into the generation result. We propose the following metrics for our tasks.

- **Vocalness** measures whether an audio clip contains singing voices. There are different publicly available tools for detecting singing voices in an audio mixture (e.g., (Lee et al., 2018)). We choose the JDC model (Kum & Nam, 2019) for it represents the state-of-the-art. In this model, the pitch contour is also predicted in addition to the vocal activation. If the pitch at a frame is outside a reasonable human pitch range (73–988 Hz defined by JDC), the pitch is set to 0 at that frame. We consider a frame as being vocal if it has a vocal activation $\geq 0.5$ AND has a pitch $> 0$. Moreover, we define the vocalness of an audio clip as the proportion of its frames that are vocal. The tool is applied to the non-silence part of an audio[7] of the generated singing voices only, excluding the accompaniment.

- **Average pitch**: We estimate the pitch (in Hz) for each frame with two pitch detection models: the state-of-the-art monophonic pitch tracker CREPE (Kim et al., 2018a), and JDC. The average pitch is computed by averaging the pitches across the frames with confidence higher than $0.5$ for CREPE, and across the frames that are estimated to be vocal for JDC.

---

[4]Please note that this is longer than the 10-second sub-clips we used to train the singer models. This is okay as our model can generate variable-length output.

[5]`https://github.com/mathigatti/midi2voice`

[6]As our models do not contain meaningful lyrics, to be fair the baselines should not contain meaningful lyrics either. We choose 'la' because people do sometimes sing with 'la' and it has no semantic meaning. An alternative way to get the lyrics is by randomly sampling a number of characters. However, randomly sampling a reasonable sequence of characters is not a trivial task as well.

[7]The non-silence frames are derived by using the `librosa` function 'effects._signal_to_frame_nonsilent.'

| Proposed model | Average pitch (Hz) | | Vocalness | Matchness |
|---|---|---|---|---|
| | **CREPE** | **JDC** | **JDC** | |
| Free singer (female) | $288 \pm 28$ | $292 \pm 28$ | $0.48 \pm 0.09$ | $-13.28 \pm 3.80$ |
| Accompanied singer (female) | $313 \pm 18$ | $316 \pm 19$ | $0.55 \pm 0.11$ | $-9.25 \pm 3.13$ |
| Solo singer (female) | $302 \pm 17$ | $306 \pm 18$ | $0.56 \pm 0.10$ | $-9.30 \pm 3.11$ |
| Free singer (male) | $248 \pm 39$ | $242 \pm 32$ | $0.44 \pm 0.16$ | $-13.29 \pm 3.19$ |
| Accompanied singer (male) | $207 \pm 14$ | $200 \pm 15$ | $0.44 \pm 0.13$ | $-9.31 \pm 3.16$ |
| Solo singer (male) | $213 \pm 14$ | $207 \pm 16$ | $0.46 \pm 0.12$ | $-9.30 \pm 3.13$ |
| **Baseline: Singing voice synthesis** | | | | |
| Sinsy (training vocal, female) | $305 \pm 59$ | $308 \pm 57$ | $0.71 \pm 0.17$ | $-9.20 \pm 3.12$ |
| Sinsy (training vocal, male) | $260 \pm 86$ | $259 \pm 72$ | $0.73 \pm 0.14$ | $-9.09 \pm 3.14$ |
| Sinsy (testing piano skyline, female) | $523 \pm 138$ | $431 \pm 62$ | $0.66 \pm 0.14$ | $-8.88 \pm 3.04$ |
| Sinsy (testing piano skyline, male) | $520 \pm 137$ | $423 \pm 61$ | $0.62 \pm 0.15$ | $-8.93 \pm 3.02$ |
| **Baseline: Training data** | | | | |
| Wikifonia: real melody-chords | — | — | — | $-7.04 \pm 2.91$ |
| Wikifonia: random melody-chords | — | — | — | $-13.16 \pm 3.72$ |
| Singer train data (vocals, female) | $312 \pm 70$ | $310 \pm 56$ | $0.60 \pm 0.14$ | $-9.24 \pm 3.09$ |
| Singer train data (vocals, male) | $263 \pm 93$ | $258 \pm 75$ | $0.64 \pm 0.16$ | $-9.09 \pm 3.22$ |
| Singer train data (accomp., female) | — | — | $0.05 \pm 0.09$ | — |
| Singer train data (accomp., male) | — | — | $0.12 \pm 0.15$ | — |
| MUSDB clean vocals | $271 \pm 81$ | $283 \pm 75$ | $0.59 \pm 0.14$ | — |

Table 1: Result of objective evaluation for our singer models and a few baseline methods.

- **Singing-accompaniment matchness**: As detailed in the appendix, to objectively measure matchness, we build a melody harmonization RNN by adapting the chord generator described in Section 3.3. Given a pair of melody and chord sequences, the model computes the likelihood of observing that chord sequence as the output when taking the melody sequence as the model input. We use the average of the log likelihood across time frames as the matchness score. As the harmonization model considers symbolic sequences, we use CREPE to transcribe the generated voices, and Madmom (Böck et al., 2016) to recognize the chord sequence from the accompaniment track.

Several observations can be made from the result shown in Table 1. In terms of the average pitch, we can see that the result of our model is fairly close to that of the singing voices in the training data. Moreover, the average pitch of the generated female voices is higher than that of the generated male voices as expected. We can also see that the Sinsy singing voices tend to have overly high pitches, when the melody line is derived from a piano playing (denoted as 'testing piano skyline.').

In terms of vocalness, our models score in general lower than Sinsy, and the singing voices in the training data. However, the difference is not that far. As a reference, we also compute the vocalness of the accompaniments in the training set (denoted as 'accomp.') and it is indeed quite low.[8]

As for matchness, we show in Table 1 the score computed from the real melody-chords pairs of Wikifonia (–7.04) and that from random pairs of Wikifonia (–13.16). We can see that the accompanied singers score higher than the random baseline and the free singer as expected.[9] Moreover, the matchenss scores of the accompanied singers are close to that of the singer training data.

Examples of the generated spectrograms of our models can be found in the appendix. From visually inspecting the spectrograms and listening to the result, the models seem to learn the characteristics of the singing melody contour (e.g., the F0 is not stable over time). Moreover, the female singer models learn better than the male counterparts, possibly because of the larger training set.

---

[8]We note that Sinsy even scores higher in vocalness than the training data. This may be due to the fact that real singing voices are recorded under different conditions and effects.

[9]The matchness scores of the free singers are computed by pairing them with the 402 test clips.

| Model (epochs trained) | Sound quality | Vocalness | Expression | Matchness |
|---|---|---|---|---|
| G3BEGAN (20 epochs) | $1.59 \pm 0.82$ | $1.93 \pm 0.99$ | $1.98 \pm 0.88$ | $2.18 \pm 1.08$ |
| G3BEGAN (240 epochs) | $2.24 \pm 0.93$ | $2.66 \pm 1.01$ | $2.60 \pm 1.01$ | $2.58 \pm 1.05$ |
| G3BEGAN (final) | $2.38 \pm 0.96$ | $2.98 \pm 1.02$ | $2.85 \pm 1.00$ | $2.74 \pm 1.04$ |

Table 2: Mean opinion scores (MOS) and standard deviations with respect to four evaluation criteria collected from the user study, for three different versions of accompanied singer (female). The scores are in 5-point Likert scale and are from 1 to 5; the higher the better.

| Model (epochs trained) | Sound quality | Vocalness | Expression | Matchness |
|---|---|---|---|---|
| G3BEGAN (final) | $1.71 \pm 0.70$ | $2.39 \pm 1.11$ | $2.27 \pm 1.06$ | $2.34 \pm 1.16$ |
| Sinsy (Oura et al., 2010) | $3.19 \pm 1.07$ | $2.90 \pm 1.01$ | $2.40 \pm 0.98$ | $2.10 \pm 0.90$ |
| Synthesizer V (Hua et al., 2019) | $3.57 \pm 1.07$ | $3.30 \pm 1.24$ | $3.25 \pm 1.10$ | $3.35 \pm 1.15$ |

Table 3: MOS from the second user study, comparing our model and two existing SVS systems.

### 4.4 User Study and Subjective Evaluation Result

We conduct two online, non-paid user studies to evaluate the accompanied singer, the female one. In the first user study, we compare the 'final' model (with the number of epochs selected according to a validation set) against two early versions of the model trained with less epochs. In the second one, we compare the proposed accompanied singer with Sinsy and Synthesizer V.

In the first study, we recruit 39 participants to each rate the generated singing for three different accompaniment tracks (each 20 seconds), one accompaniment track per page. The subjects are informed the purpose of our research (i.e., score and lyrics-free singing voice generation) and the user study (to compare three computer models), and are asked to listen in a quiet environment with proper headphone volume. No post-processing (e.g., noise removal, EQ adjustment) is applied to the audio. The ordering of the result of the three models is randomized.

The process of the second study is similar to the first one, but it includes five different accompaniments (randomly chosen from those used in the first user study) and the respective generated/synthesized singing voices. The melodies used for synthesis are those from the piano skyline of the test data, so that our model can be compared with the synthesis methods with the same accompaniment. A separate set of 21 subjects participate in this study. The audio files used in this user study can be downloaded from `https://bit.ly/2qNrekv`.

Tables 2 and 3 show the result of the two studies. We can see that the model indeed learns better with more epochs. Among the four evaluation criteria, the Sound Quality is rated lower than the other three in both studies, suggesting room for improvement. By comparing the proposed model with the two SVS systems, we see that Synthesizer V performs the best for all the evaluation criteria. Our model achieves better Matchness than Sinsy, and achieves a rating close to Sinsy in Expression.[10] In general, we consider the result as promising considering that our models are trained from scratch with little knowledge of human language.

## 5 Related work

While early work on SVS is mainly based on digital signal processing (DSP) techniques such as sampling concatenation (Cook, 1996; Bonada & Serra, 2007), machine learning approaches offer greater flexibility and have been more widely studied in recent years. Hidden Markov models

---

[10]We note that Sinsy and Synthesizer V have an unfair advantage on matchness because their singing voices are basically synthesized according to the melody lines of the accompaniment. From Table 3, we see that Synthesizer V does exhibit this advantage, while Sinsy does not. We observe that the Sinsy singing voices do not always align with the provided scores. The fact that Synthesizer V has higher audio quality seem to promote its score in the other criteria. The presence of the result of Synthesizer V seems to also make the subjects in the second study rate the proposed model lower than the subjects do in the first study.

(HMMs), in particular, have been shown to work well for the task (Saino et al., 2006). The Sinsy system, a baseline model in Section 4, is also based on HMMs (Oura et al., 2010). Nishimura et al. (2016) report improved naturalness by using deep neural nets instead of HMMs. Since then, many neural network models have been proposed.

The model presented by Nishimura et al. (2016) uses simple fully-connected layers to map symbolic features extracted from the user-provided scores and lyrics, to a vector of acoustic features for synthesis. The input and output features are time-aligned frame-by-frame beforehand by well-trained HMMs. The input features consist of score-related features (e.g., the key of the current bar and the pitch of the current musical note), and lyrics-related ones (the current phoneme identify, the number of phonemes in the current syllable, and the duration of the current phoneme). The output features consist of spectral and excitation parameters and their dynamic features (Hono et al., 2018), which altogether can then be turned into audio with a DSP technique called the MLSA filter (Imai, 1983).

The aforementioned model has been extended in many aspects. For instance, using convolutional layers and recurrent layers in replacement of the fully-connected layers for learning the mapping between input and output features has been respectively investigated by Nakamura et al. (2019) and Kim et al. (2018b). Using neural vocoders such as the WaveNet (van den Oord et al., 2016) instead of the MLSA filter has been shown to improve naturalness by Nakamura et al. (2019). Rather than using hand-crafted features for the input and output, Lee et al. (2019a) train a model to predict the mel-spectrogram directly from time-aligned lyrics and pitch labels, and then use the Griffin-Lim algorithm (Griffin & Lim, 1984) to synthesize the audio. Modern techniques such as adversarial loss and attention module have also been employed (Lee et al., 2019a). A follow-up work adds a speaker encoder to the network to achieve multi-singer SVS (Lee et al., 2019b). Synthesizer V (Hua et al., 2019), the other baseline model we employ in Section 4, is based on a hybrid structure that uses both deep learning and sample-based concatenation.[11]

While exciting progress has been made to SVS, the case of score and lyrics-free singing voice generation, to our best knowledge, has not been tackled thus far. Similar to (Lee et al., 2019a), we do not use hand-crafted features and we train our model to predict the mel-spectrograms.

Using neural nets for score-conditioned instrumental audio generation have also been investigated in recent years. However, existing work is mostly concerned with the generation of single notes of, for example, 4-second long (Défossez et al., 2018; Donahue et al., 2019; Engel et al., 2019). A deep neural network that is capable of generating variable-length audio (e.g., a "recurrent generator") as the proposed singer models do, to our knowledge, has not been much studied.

## 6 CONCLUSION

In this paper, we have introduced a novel task of singing voice generation that does not use musical scores and lyrics. Specifically, we proposed three singing schemes with different input conditions: free singer, accompanied singer, and solo singer. We have also proposed a BEGAN based architecture that uses GRUs and grouped dilated convolutions to learn to generate singing voices in an adversarial way. For evaluating such models, we proposed several objective metrics and implemented a model to measure the compatibility between a given accompaniment track and the generated vocal track. The evaluation shows that the audio quality of the generated voices still leave much room for improvement, but in terms of humanness and emotion expression our models work fine.

Score and lyrics-free singing voice generation is a new task, and this work represents only a first step tackling it. There are many interesting ideas to pursue. For example, we have chosen to extract pitch-related information only from the accompaniment track for the accompanied singer, but a more interesting way is to let the model learns to extract relevant information itself. In the near future, we plan to investigate advanced settings that allow for timbre and expression control, and experiment with other network architectures, such as coupling a fine-grained auto-regressive model with a multiscale generation procedure as done in MelNet (Vasquez & Lewis, 2019), using a discriminator that examines different chunks of the generated audio as done in PatchGAN for the vision domain (Isola et al., 2017), or using multiple discriminators that evaluate the generated audio based on multi-frequency random windows as done in GAN-TTS (Bińkowski et al., 2019).

---

[11]https://synthv.fandom.com/wiki/File:Synthesizer_V_at_the_Forefront_of_Singing_Synth (last accessed: Nov. 12, 2019)

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

## A    APPENDIX: TRAINING SETTING AND MODEL DETAILS

### A.1    G3BEGAN

Singer models with G3BEGAN are trained with Adam (Kingma & Ba, 2014) with 0.0001 learning rate and mini-batch size being 5. Each model is trained with 500 epochs. Gradient norm clipping with magnitude 3 is used.

The generator in G3BEGAN is implemented with a stack of two G3 blocks. Please see Table 4 for details of the network architecture.

### A.2    CHORD GENERATOR

The chord generator is aimed to generate chord progressions freely under some given conditions. It supports 12 major and 12 minor keys, 10 tempo options from 60 to 240 BPM, 6 time signature options, and 51 chord qualities (612 chords in total). The conditions, key, tempo, and time signatures, are encoded into one-hot representation and concatenated together as a 40-dimension vector. The model mainly consists with 3 stacked GRU layers, each with 512 hidden variables. The input of each time step is a 524-dimensional vector consisting of a chord embedding and a beat-related one-hot positional encoding (to encourage the model to follow certain rhythmical pattern. This input array passes through a fully-connected layer to 512-dimension and is used as the input of the GRUs. The training data are the leadsheets from the Wikifonia dataset. We augmented the data by rotating the keys, leading to in total 80,040 leadsheets for training.

| | Details | Input dim | Output dim |
|---|---|---|---|
| **Input** | 1DConv (kernel=3, dilation=1)
Group normalization (group=4)
Leaky ReLU (0.01) | | 512 |
| **G3 Block 1** | GRU
Grouped 1DConv (kernel=3, dilation=2, group=4)
Group normalization (group=4)
Leaky ReLU (0.01) | 512
512 | 512
512 |
| **G3 Block 2** | GRU
Grouped 1DConv (kernel=3, dilation=2, group=4)
Group normalization (group=4)
Leaky ReLU (0.01) | 512
512 | 512
512 |
| **Output** | 1DConv (kernel=3, dilation=1) | 512 | 80 |

Table 4: Network architecture of the generator (G) and the discriminator (D) of the proposed G3BEGAN model. It uses two G3 blocks introduced in Section 3.1. For the noise input, we set the dimension $U$ to 20. For the G and D of a free singer, the input dimensions are 20 and 80, respectively. For the G and D of an accompanied singer, the input dimensions are $100 = 20 + 80$ and $160 = 80 + 80$, respectively.

| | Usage | Processing | Quantity |
|---|---|---|---|
| Singer training data (female) | Tr. female singers | SS, PT | 9.9 hrs |
| Singer training data (male) | Tr. male singers | SS, PT | 5.0 hrs |
| Jamendo test data | Ev. singers | SS, PT | 5.3 hrs |
| Vocoder training data | Tr. WaveRNN vocoder | SS | 3,500 tracks |
| MAESTRO | Tr. piano transcription | | 200 hrs |
| MUSDB + 4.5-hr Piano Solo | Tr. source separation model | | 100 tracks + 4.5 hrs |

Table 5: A summary of the datasets employed in this work. The first three datasets and the 4.5-hour Piano solo part of the last dataset are Jazz music, whereas the others may not. The 'Processing' column indicates the music processing models that have been applied to process the respective dataset. (Notation—Tr.: training. Ev.: evaluating. SS: source separation. PT: piano transcription)

### A.3 MELODY HARMONIZATION MODEL

The melody harmonization model is modified from the chord generator described in Appendix A.2, using additionally the melody tracks found in the Wikifonia dataset. Specifically, the model intends to generate a chord sequence given a melody sequence. Such a model can be learned by using the pairs of melody and chord tracks in Wikifonia. We add the chroma representation of the melody with window size of a quarter-note to the input vector.

The matchness of the real pairs of melody and chord progression in Wikifonia is –7.04±2.91. If we pair a melody with a randomly selected chord progression and calculate the matchness, the score becomes –13.16±3.72.

## B APPENDIX: DATASETS

## C APPENDIX: EXAMPLES OF THE SPECTROGRAM OF THE GENERATED SINGING VOICES

Examples of the spectrograms of the generated singing voices can be found in Figures 3 and 4.

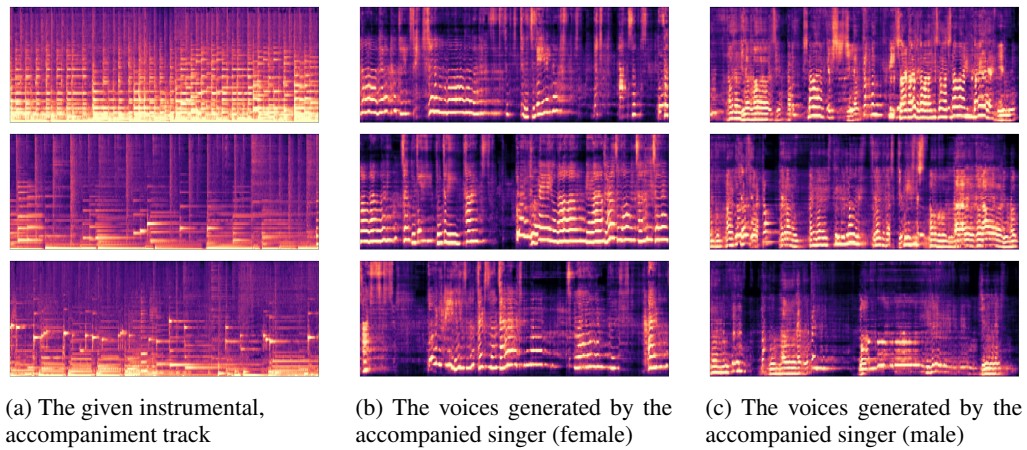

(a) The given instrumental, accompaniment track

(b) The voices generated by the accompanied singer (female)

(c) The voices generated by the accompanied singer (male)

Figure 3: Samples of spectrograms generated by our accompanied singers: (left) the given accompaniment tracks, the voices generated by (middle) the female singer and (right) the male singer.

| Model | Average pitch (Hz) | | Vocalness |
|---|---|---|---|
| | CREPE | JDC | JDC |
| BEGAN (female) | $288 \pm 28$ | $292 \pm 28$ | $0.48 \pm 0.09$ |
| GAN (female) | $307 \pm 7$ | $371 \pm 25$ | $0.22 \pm 0.17$ |
| LSGAN (female) | $1130 \pm 9$ | $453 \pm 14$ | $0.28 \pm 0.03$ |

Table 6: Result of using different GAN losses, for free singer (female).

## D    APPENDIX: ABLATION STUDY ON DIFFERENT GAN LOSSES

We experiment different GANs, including BEGAN (Berthelot et al., 2017), vanilla GAN (Goodfellow et al., 2014), and LSGAN (Mao et al., 2017), for the case of building the free singer. The output of the discriminators in the vanilla GAN and LSGAN is a single real/fake value. To compare the three GANs as fairly as possible, the discriminators used in vanilla GAN and LSGAN are almost the same as the one used in the BEGAN. The only difference is that the discriminators used in vanilla GAN and LSGAN have an extra average-pooling layer in the output. The validation losses at different training epochs are shown in Figure 5, and the metrics are shown in Table 6. In Figure 5, we can see that, according to our implementation, only BEGAN converges. In Table 6, we can see that the BEGAN model has much higher Vocalness than other models. By listening to the generated singing voices of the GAN and LSGAN models, we find that they are basically noises.

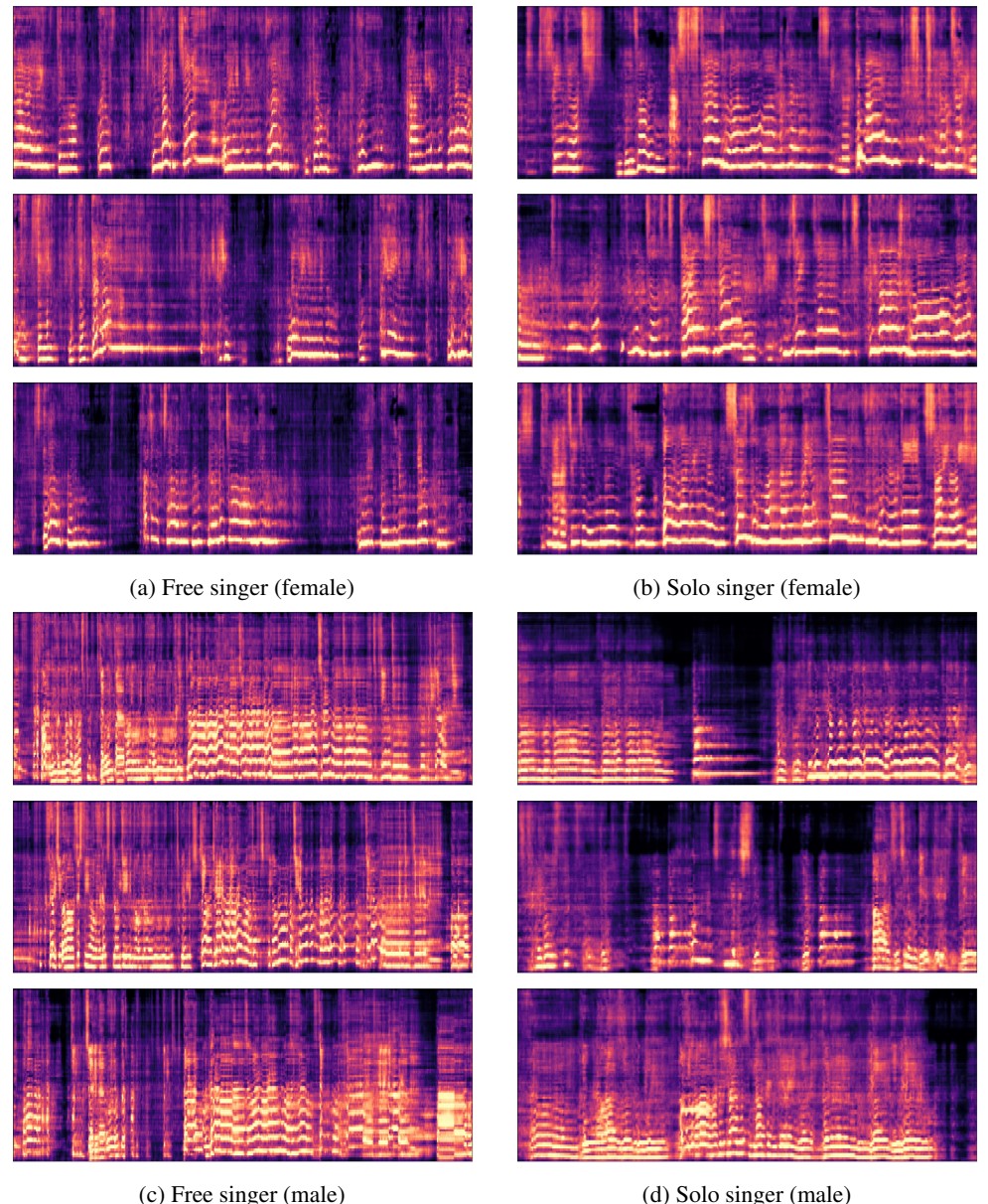

Figure 4: Samples of spectrograms generated by our (left) free singers and (right) solo singers. We can see salient pitch contour in the spectrograms. Moreover, the pitches sung by the male singers seem on average lower than those sung by the female singers.

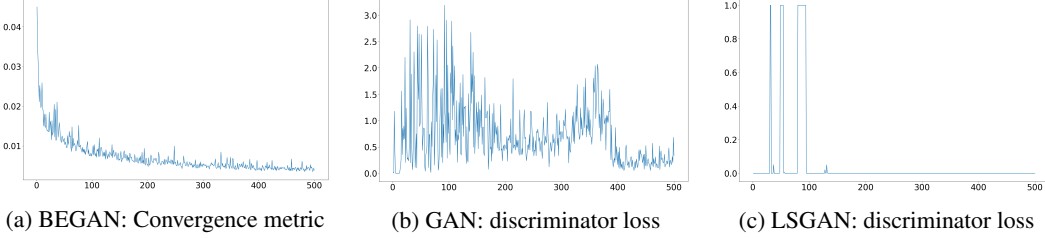

Figure 5: Loss trends of different GAN losses. BEGAN comes with a convergence metric (Berthelot et al., 2017) which can be used to examine how well the model converges.