# OpenReview forum: "Score and Lyrics-Free Singing Voice Generation"
_ICLR.cc/2020/Conference — Reject_

### Official Review · AnonReviewer1 · 2019-10-26
**Official Blind Review #1**

**Rating:** 3

**Review:**

In this paper, authors explore the problem of generating singing voice, in the waveform domain. There exists commercial products which can generate high fidelity sounds when conditioned on a score and or lyrics. This paper proposes three different pipelines which can generate singing voices without necessitating to condition on lyrics or score.

Overall, I think that they do a good job in generating vocal like sounds, but to me it's not entirely clear whether the proposed way of generating melody waveforms is an overkill or not. There is a good amount of literature on generating MIDI representations. One can simply generate MIDI (conditioned or unconditioned), and then give the result to a vocaloid like software. I am voting for a weak rejection as there is no comparison with any baseline. If you can provide a comparison with a MIDI based generation baseline, I can reconsider my decision. Or, explain to me why training on raw waveforms like you do is more preferable. I think in the waveform domain may even be undesirable to work with, as you said you needed to do source separation, before you can even use the training data. This problem does not exist in MIDI for instance.


**Experience Assessment:**

I have read many papers in this area.

**Review Assessment: Checking Correctness Of Derivations And Theory:**

N/A

**Review Assessment: Checking Correctness Of Experiments:**

I assessed the sensibility of the experiments.

**Review Assessment: Thoroughness In Paper Reading:**

I read the paper at least twice and used my best judgement in assessing the paper.

---

> ### Author Response · Authors · 2019-11-13
> **Response to the Review # 1**
>
> Thank you for the valuable comments. We address the issues and questions raised by the reviewer in the following comments.
>
> $\textbf{1.}$ There is a good amount of literature on generating MIDI representations. One can simply generate MIDI (conditioned or unconditioned), and then give the result to a vocaloid like software. I am voting for a weak rejection as there is no comparison with any baseline. If you can provide a comparison with a MIDI based generation baseline, I can reconsider my decision.
>
> $\textbf{Ans:}$
> One of the goals in this paper is to generate singing voice given an accompaniment, but generating a singing melody MIDI given an accompaniment is not a trivial task. To the best of our knowledge, there are many researches on generating melodies, piano solos, and scores of several instruments, but very few researches, if any, work on generating singing melodies given an accompaniment. Our model is one way to achieve the goal of generating singing given accompaniment without the intermediate MIDI file.
>
> As suggested by the Reviewer 1, we have added two synthesis baselines to the paper, and have revised Section 4 accordingly. The two baselines are based on the well-known singing voice synthesis tools, Sinsy and Synthesizer V, that are publicly accessible.
>
>
> $\textbf{2.}$ Or, explain to me why training on raw waveforms like you do is more preferable. I think in the waveform domain may even be undesirable to work with, as you said you needed to do source separation, before you can even use the training data. This problem does not exist in MIDI for instance.
>
> $\textbf{Ans:}$
> We believe that our score-lyrics-free approach and the score-lyrics-based (with both score and lyrics) approach are for different situations, so one approach is not preferable than the other in general.
>
> The differences between these two approaches is in the types of the input conditions. The score-lyrics-based approach takes scores and lyrics as the condition. In contrast, our approach can take less strict and more diverse conditions. In this paper, we have demonstrated models that take no conditions, accompaniment conditions, or chord conditions.
>
> What we propose in this paper is that the conditions used in the singing voice generation systems do not need to be as strong as the existing systems. Score-lyrics-based systems are good at synthesizing singing voices when you want the system to synthesize exactly what you want, while our approach would do better when you want the machine to add some singing voices to your composition without designating the strict scores and lyrics.
>
> We have also revised the Introduction to further discuss the motivations of using our approaches.

---

### Official Review · AnonReviewer5 · 2019-11-06
**Official Blind Review #5**

**Rating:** 1

**Review:**

This paper has set a new problem: singing voice synthesis without any score/lyrics supervision. The authors provide a significance of such a problem in section 1. Also, the authors successfully design and implement a novel neural network architecture to solve the problem. It’s also notable that the authors kindly open-source their code to mitigate the reproducibility issue. This paper may serve as baseline results for the proposed problem in the future.

Despite the significance of the problem and the novelty of the solution, this paper aims to solve too many problems at once. Unfortunately, some main ideas were not supported by experimental results or logical arguments with appropriate citations.

The authors seem to overly focus on the task itself, and thus haven’t pay much attention on supporting their choice of neural network architecture. Here are some points regarding this:

1. “We adapt the objective of BEGAN, which is originally for generating images, for generating sequences.”: The original BEGAN paper(Berthelot et al., 2017) did not address sequence modeling.
2. “As for the loss function of D(·), our pilot experiments show that …”: This hand-wavy argument is unacceptable. The authors should be able to support all of the claims they’ve made, which sometimes require experimental results. “G(·)” of the following sentence should be D(·).
3. “Our conjecture is that, … compressing the output of G(·) may have lost information important to the task.”: PatchGAN (Isola et al., 2017) had already addressed this issue. The authors may want to cite PatchGAN to support their conjecture or compare against PatchGAN to show their own architecture’s strength.
4. “We like to investigate whether such blocks can help generating plausible singing voices in an unsupervised way.”: No ablation studies on GRUs and dilated convolutions are found. If the authors mean that they’re willing to do such studies in the future, “what’s done here” and “what will be done in the future” should be easily distinguished within the text.

Some miscellaneous points worth noting:
1. The readers won’t be able to estimate the strength of the proposed method by looking at table 1 and 2. I suggest doing one of the following: include results from other baselines to compare against or give a brief description of the metrics with typical values. (e.g. values shown in appendix A.3)
2. Are the neural network architecture components described in section 3.1 used for both source separation and the synthesis network?
3. To make readers easily understand the contribution of this paper, there should be a detailed description of the limitation of this work. I suggest to move the details of experiments in section 4 to the appendix, but it may depend on the authors’ writing style.
4. The ‘inner idea’ concept in the “solo singer” setting looks vague and contradicts with the main topic since it uses chord sequences to synthesize singing voice.

Things to improve the paper that did not impact the score:
1. “improv” >> “improvement.”

This paper should be rejected because (1) the paper failed to justify the main idea and results, (2) the amount of literature research was not enough, (3) too many problems were addressed at once, and (4) the writing is not clear enough.

**Experience Assessment:**

I have read many papers in this area.

**Review Assessment: Checking Correctness Of Derivations And Theory:**

I assessed the sensibility of the derivations and theory.

**Review Assessment: Checking Correctness Of Experiments:**

I did not assess the experiments.

**Review Assessment: Thoroughness In Paper Reading:**

I made a quick assessment of this paper.

---

> ### Author Response · Authors · 2019-11-13
> **Response to the Review #5**
>
> Thank you for the valuable comments. We address the issues and questions raised by the reviewer in the following comments.
>
> $\textbf{A.1.}$ “We adapt the objective of BEGAN, which is originally for generating images, for generating sequences.”: The original BEGAN paper(Berthelot et al., 2017) did not address sequence modeling.
>
> $\textbf{Ans:}$
> Yes, with that sentence we did mean that "BEGAN is originally proposed for generating images, not sequences. Therefore, to make it generate sequences, we have modified/adapted the objective of the BEGAN model".   We realize that the sentence (in the original version of our submission) may be confusing.  To avoid misunderstanding, we have revised the whole paragraph for clarification.
>
>
> $\textbf{A.2.}$ “As for the loss function of D(·), our pilot experiments show that …”: This hand-wavy argument is unacceptable. The authors should be able to support all of the claims they’ve made, which sometimes require experimental results.
>
> $\textbf{Ans:}$
> We have added a comparison of BEGAN, GAN, and LSGAN in Appendix D. It shows that BEGAN achieves the best convergence. Furthermore, we also show the objective metrics of the three Gan losses, and see that BEGAN achieves the best vocalness among the three GAN losses in Table 6.
>
>
> $\textbf{A.3.}$ “Our conjecture is that, … compressing the output of G(·) may have lost information important to the task.”: PatchGAN (Isola et al., 2017) had already addressed this issue. The authors may want to cite PatchGAN to support their conjecture or compare against PatchGAN to show their own architecture’s strength.
>
> $\textbf{Ans:}$
> We have revised the paragraph of “Our conjecture is that...” for clarification and the sentence has been removed. However, after reading the paper, we do think that the approach of PatchGAN does share some similar rationale with BEGAN, so we have referred to it in the conclusion as a possible direction for the future work.
>
>
> $\textbf{A.4.}$ “We like to investigate whether such blocks can help generating plausible singing voices in an unsupervised way.”: No ablation studies on GRUs and dilated convolutions are found. If the authors mean that they’re willing to do such studies in the future, “what’s done here” and “what will be done in the future” should be easily distinguished within the text.
>
> $\textbf{Ans:}$
> We have rephrased it as follows:
> For it has demonstrated its capability in source separation, we adopt it as a building block of the singer models.
>
>
> $\textbf{B.1.}$ The readers won’t be able to estimate the strength of the proposed method by looking at table 1 and 2. I suggest doing one of the following: include results from other baselines to compare against or give a brief description of the metrics with typical values. (e.g. values shown in appendix A.3)
>
> $\textbf{Ans:}$
> We have included both of them in the revised version.
> In Section 4, two synthesis baselines with Sinsy and Synthesizer V are included, and the typical values computed on the training data are provided.
>
>
> $\textbf{B.2.}$ Are the neural network architecture components described in section 3.1 used for both source separation and the synthesis network?
>
> $\textbf{Ans:}$
> For the source separation, we use the original design by (Liu & Yang, 2019) with weight normalization for convolution layers. For the generation network, we replace the weight normalization with group normalization for convolution layers.
>
>
> $\textbf{B.3.}$ To make readers easily understand the contribution of this paper, there should be a detailed description of the limitation of this work. I suggest to move the details of experiments in section 4 to the appendix, but it may depend on the authors’ writing style.
>
> $\textbf{Ans:}$
> We have revised Introduction so that the motivation and the scope of this work should be clearer in the revised version.
>
>
> $\textbf{B.4.}$ The ‘inner idea’ concept in the “solo singer” setting looks vague and contradicts with the main topic since it uses chord sequences to synthesize singing voice.
>
> $\textbf{Ans:}$
> By score-free we mean that the model is not asked to strictly follow a pre-assigned score that contains pitch information. In the solo singer, the generated singing voices are not asked to follow the pitch notes in the chord sequence, so we still consider it to be a score-free approach.
>
> The general “inner idea” concept is indeed vague in the paper, but it is vague because there could be many options of it. In the paper, we instantiate the inner idea by setting it to be a chord sequence, which provides the readers an example of what it could be and hopefully make it less vague.
>
>
> $\textbf{B.5.}$ Things to improve the paper that did not impact the score:
> 1. “improv” >> “improvement.”
>
>
> $\textbf{Ans:}$
> Thank you for pointing this out. We have corrected it.

---

### Official Review · AnonReviewer4 · 2019-11-06
**Official Blind Review #4**

**Rating:** 3

**Review:**

This paper tries to addresses an interesting problem of generating singing voice track under three different circumstances. Some of the problems that this paper deals with is a new problem and introduced first in this paper, which could be a contribution as well.

Although the paper is fairly well-structured and written, especially in the early sections, I am giving a weak reject due to its weak evaluation. Evaluation is almost always difficult when it comes to generative art, but I am slightly more concerned than that. The literature review can be, although it is nicely done overall, improved, especially on the neural network based singing voice synthesis.

I appreciate the authors tried to find a great problem and provided a good summary of the literature. Successfully training this kind of network itself is already tricky. It is also nice to see some interesting approaches towards objective evaluation.

Below are my comments.

> Our conjecture is that, as the output of G(·) is a sequence of vectors of variable length (rather than a fixed-size image), compressing the output of G(·) may have lost information important to the task.

I am rather not convinced. The difficulty to discriminate them doesn't seem to be (strongly) related to their variable length for me because a lot of papers have, at least indirectly, dealt with such a case successfully.

> For data augmentation, we transpose the chord progressions found in Wikifonia to 24 possible keys

What do you mean by 24 keys? I think there should be only 12 keys.

> Vocalness measures whether a generated singing voice audio sounds like vocals. We use the singing voice detection tool proposed by Leglaive et al. (2015) and made available by Lee et al.(2018).

Actually, the paper (Lee et al. 2018) suggested that vocal activity detection in the spectrum domain is easily affected by some features such as frequency modulation. I am not sure if this feature is suitable as a measure of the proposed vocalness. The computed vocalness may provide more information if they are computed on other tracks (e.g., guitar, cello, drums, etc).

> Average pitch: We use the state-of-the-art monophonic pitch tracker CREPE (Kim et al., 2018)8 to compute the pitch (in Hz) for each frame. The average pitch is computed by averaging the pitches of all the frames.

I am pretty sure that this is not the right way to evaluate as a metric of generated vocal track. CREPE is a neural network based pitch tracker, which means it is probably biased by the training data, where the pitch values mostly range in that of common musical instruments/voices. This means, when the F0 of input is not really in the right range, CREPE might incorrectly predict somewhat random F0 within THAT range anyway. I'd say the distribution of pitch values can be interesting metric to show and discuss, but not as a metric of vocal track generation.





**Experience Assessment:**

I have published one or two papers in this area.

**Review Assessment: Checking Correctness Of Derivations And Theory:**

I did not assess the derivations or theory.

**Review Assessment: Checking Correctness Of Experiments:**

I assessed the sensibility of the experiments.

**Review Assessment: Thoroughness In Paper Reading:**

I read the paper at least twice and used my best judgement in assessing the paper.

---

> ### Author Response · Authors · 2019-11-13
> **Response to the Review #4**
>
> Thank you for the valuable comments. We address the issues and questions raised by the reviewer in the following comments.
>
>
> $\textbf{1.}$ Although the paper is fairly well-structured and written, especially in the early sections, I am giving a weak reject due to its weak evaluation.
>
> $\textbf{Ans:}$
> Thank you for the comment. That is also the concern of all the reviewers. We have largely expanded the evaluation to include two types of baselines.
> a. We have included two baseline systems of singing voice synthesis: Sinsy and Synthesizer V.
> b. We have also computed the metrics on the training data, so that there are more references for comparison.
>
>
> $\textbf{2.}$ The literature review can be, although it is nicely done overall, improved, especially on the neural network based singing voice synthesis.
>
> $\textbf{Ans:}$
> We have expanded literature review to include more neural network based methods in Section 5.
>
>
> $\textbf{3.}$ Our conjecture is that, as the output of G(·) is a sequence of vectors of variable length (rather than a fixed-size image), compressing the output of G(·) may have lost information important to the task.
>
> I am rather not convinced. The difficulty to discriminate them doesn't seem to be (strongly) related to their variable length for me because a lot of papers have, at least indirectly, dealt with such a case successfully.
>
> $\textbf{Ans:}$
> You are right, and we did not mean that the failure of training with the vanilla GAN loss is due to the variable-length sequences in our training set. In fact, in the training phase, we use fixed-length sequences. What we want to express is that the compression of a sequence (whether it is variable-length or not) into a single true/false value could be a cause for the failure.
>
> We have revised the whole paragraph to clarify the description. We have also added a comparison of training with GAN, LSGAN, and BEGAN in Appendix D.
>
>
> $\textbf{4.}$ Regarding the transposing of Wikifonia chord progressions
>
> $\textbf{Ans:}$
> Yes, there are only 12 keys. Thank you for spotting the typo. We have revised it.
>
>
> $\textbf{5.}$ the paper (Lee et al. 2018) suggested that vocal activity detection in the spectrum domain is easily affected by some features such as frequency modulation. I am not sure if this feature is suitable as a measure of the proposed vocalness.
>
> $\textbf{Ans:}$
> Thank you very much for this comment. Indeed, (Lee et al. 2018) shows that the frequency modulation is an important factor causing high false positive rates when some types of instruments are present in the songs. However, we compute the vocalness measures on the non-silence part of the generated singing voices only, without the accompaniment, so the effect of frequency modulation might not be as serious as the scenario investigated in (Lee et al. 2018).
>
> On the other hand, we also agree that there are better ways to compute the vocalness, so we have devised a way to compute the vocalness taking into account both the vocal activation and the singing pitch range.
>
> For the new vocalness, we use the JDC model (https://github.com/keums/melodyExtraction_JDC) for it represents the state-of-the-art. In this model, the pitch contour is also predicted in addition to the vocal activation. If the pitch at a frame is outside a reasonable human pitch range (73~988 Hz defined by JDC), the pitch is set to 0 at that frame. We consider a frame as being vocal if it has a vocal activation >= 0.5 AND has a pitch >0. Moreover, we define the vocalness of an audio clip as the proportion of its frames that are vocal. The tool is applied to the non-silence part of an audio.
>
> We have revised the paper for this modification.
>
>
> $\textbf{6.}$ Average pitch: We use the state-of-the-art monophonic pitch tracker CREPE (Kim et al., 2018) ...
>
> I am pretty sure that this is not the right way to evaluate as a metric of generated vocal track. CREPE is a neural network based pitch tracker, which means it is probably biased by the training data, .... This means, when the F0 of input is not really in the right range, CREPE might incorrectly predict somewhat random F0 within THAT range anyway. I'd say the distribution of pitch values can be interesting metric to show and discuss, but not as a metric of vocal track generation.
>
> $\textbf{Ans:}$
> First of all, we have corrected our description of the “Average pitch” to clarify that the “Average pitch” is computed over all the frames with confidence value >= 0.5, not over all the frames. Second, we agree that there still could be incorrectly predicted F0 due to the bias of the data used to train CREPE. In our evaluation in Section 4.3, we use them to show that the two models trained with female vocals and male vocals do exhibit different characterizations in pitch, not as an absolute metric, so we think it is still a valuable metric.
>
> Furthermore, we add another way of computing average pitch by using JDC, where vocalness is also taken into account to filter out non-vocal frames.

---

> > ### Comment · AnonReviewer4 · 2019-11-14
> > **Thanks for the improvement**
> >
> > Hi,
> > I appreciate the authors have tried to address the issues raised by me and other reviewers. It's not an easy decision but I will keep my original decision.
> >
> > But maybe what's more important is the feedback reviews can provide. At least I want to believe so :) I think the added comments on evaluation metrics are helpful. However, that does not change the question of if those are approximately good ways to measure the quality of the generated samples. This is partly due to the fact that it's a new problem, but, at the end, I'd expect the authors who are suggesting a new problem would also bring up a nice initial approach to evaluate a solution. Unfortunately, I don't think this was the case. The vocalness and the average pitch does not seem to be able to assess any scenario. This work is not about simply generating a plausible voice or vocal track - it is about *singing*. Those metrics hardly consider any aspect of singing.
> >
> > I downloaded the Google Drive file but it was not very clear to figure out quickly which files are generated in which way and what aspects are expected to be evaluated precisely.

---

> > > ### Author Response · Authors · 2019-11-15
> > > **Regarding the evaluation metrics**
> > >
> > > Thank you very much for your comments. As you stated in the two review comments, the evaluation of this task is difficult because it is generative art and it is a new task. Furthermore, we would like to emphasize that it is difficult also because the evaluation of singing as a type of music is fairly subjective.
> > >
> > > For  such a subjective task, the objective metrics could at best evaluate those that can be more objectively measured. With Average Pitch, Vocalness, and Matchness, we attempt to evaluate pitch, timbre, and harmonization, respectively, which are three important aspects of singing/music.
> > >
> > > Admittedly, these objective metrics cannot fully evaluate the generated singing voices. We attempt to alleviate this situation by conducting the subjective evaluation through the two MOS user studies. These user studies include questions regarding Sound quality and Expression that could complement the objective metrics.
> > >
> > > Regarding the audio files used in the user study, each folder in the zip file contains a set of audios corresponding to one 20-second accompaniment. The set of audios include:
> > > _accompaniment.mp3: the accompaniment audio
> > > my_singer.mp3: the audio generated by our accompanied singer
> > > sinsy.mp3: the audio synthesized with Sinsy
> > > synthesizerV.mp3: the audio synthesized with Synthesizer V

---

### Official Review · AnonReviewer6 · 2019-11-06
**Official Blind Review #6**

**Rating:** 3

**Review:**

This paper claims to be the first to tackle unconditional singing voice generation. It is noted that previous singing voice generation approaches leverage explicit pitch information (either of an accompaniment via a score or for the voice itself), and/or specified lyrics the voice should sing. The authors first create their own dataset of singing voice data with accompaniments, then use a GAN to generate singing voice waveforms in three different settings:
1) Free singer - only noise as input, completely unconditional singing sampling
2) Accompanied singer - Providing the accompaniment *waveform* (not symbolic data like a score - the model needs to learn how to transcribe to use this information) as a condition for the singing voice
3) Solo singer - The same setting as 1 but the model first generates an accompaniment then, from that, generates singing voice

Firstly, the authors have done a lot of work - first making their own data, then designing their tasks and evaluating them. The motivation is slightly lacking - it is not clear why we are interested in these three task settings i.e. what we will learn from a difference in their performance, and there is a lack of discussion about which setting makes for better singing voice generation. Also, there is no comparison with other methods: whilst score data is not available it could be estimated, then used for existing models, providing a nice baseline e.g. first a score is extracted with a state of the art AMT method, then a state of the art score to singing voice generation method could be used.

There are existing datasets of clean singing voice and accompaniment, for example MIR-1k (unfortunately I think iKala, another dataset, is now unavailable). It is true that this dataset is small in comparison to the training data the authors generate, but it will certainly be cleaner. I would have liked to see an evaluation performed on this data as opposed to another dataset which was the result of source separation (the authors generate a held out test set on Jazz from Jamendo, on which they perform singing voice separation).

I also had questions about the training data - there is very little information about it included other than it is in-house and covers diverse musical genres (page 6 under 4.1), a second set of 4.5 hours of solo piano, and a third set (?) of jazz singers. This was a bit confusing and could do with clarification. At minimum, I would like to know what genre we are restricting ourselves to - is everything Jazz? Are the accompaniments exclusively piano (it's alluded that the answer to this is no, but it's not clear to me)? Is there any difference between the training and test domain.

On page 6, second to last paragraph when discussing the validation set, I would like the sampling method to be specified - it makes a difference whether the same piece of music will be contained within both the training and validation split, or whether the source piece (from which the 10 second clips are extracted) are in separate splits <- I'd recommend that setting.

The data used to train the model will greatly affect my qualitative assessment of the provided audio samples so, without a clear statement on the training data used, I can't really assess this.

However, with respect to the provided audio samples, I'd first note that these are explicitly specified as randomly sampled, and not cherry picked, which is great, thank you. However, whilst I would admit that the domain is different, when the singing samples are compared with the piano generation unconditional samples of MelNet (https://audio-samples.github.io/#section-3), which I would argue is just as hard to make harmonically valid, they are not as harmonically consistent, even when an accompaniment has been provided. However, samples do sound like human voice, and the pitch is relatively good. The words are unintelligible, but this is explicitly specified as out of scope for this paper, and I agree that this is much harder to achieve.

As an aside, MelNet is not cited in this paper and, given the similarity and relevance, I think it probably should be - https://arxiv.org/abs/1906.01083. It was published this year however so it would be a little harsh to expect it to be there. I would invite the authors to rebut this claim if they think the methods are not comparable.

My main criticism is in relation to the evaluation. For Table 2, without a baseline or the raw data (which would have required no further effort) included in the MOS study, it's very difficult to judge success. If the authors think that comparison with raw data is unfair (as it is an embryonic task) they could include a model which has an unfair advantage from the literature - e.g. uses extracted score information.

For Table 1, I appreciate the effort that went into the design of 'Vocalness' and 'Matchness' which are 'Inception Score' type metrics leaning on other learned models to return scores. I would like to see discussion which explains the differences in scores for the different model settings (there is a short sentence at the bottom of page 7, but nothing on vocalness).

In summary - this is a hard problem and the authors are the first to tackle it. The different approaches to solve the problem are not well motivated. However, the models are detailed, and well explained. Code is even provided, but data for training is not. If the authors were able to compare with a baseline (like that I describe above), it would go a long way to convincing me that this was good work. As it stands, Tables 1 and 2, and the provided audio samples have no context, so I cant make a conclusion. If this issue and motivation was addressed I would likely vote to accept the paper.

Things to improve the paper that did not impact the score:
1. p2 "we hardly provide any labelled data" specify whether you do or not (I think it's entirely unsupervised since you extract chord progressions and pitch curves using learned models...)
2. p2 "...may suffer from the artifact" -> the artefacts
3. p2 "for the scenario addressed by the accompanied singer" a bit clumsy, may be worth naming your tasks 1, 2 and 3 such that you can easily refer to them
4. p2 "We investigate using conditional GAN ... to address this issue" - which issue do you mean? If it is the issue specified at the top of the paragraph, i.e. that there are many valid melodies for a given harmony (no single ground truth), I don't think using a GAN is a *solution* to this per se. It is a valid model to use, and the solution would be enough varied data (and evaluation to show you're covering your data space and haven't collapsed to a few modes)
5. p2 "the is no established ways" -> there are no established ways
6. p3 "Discriminators in GAN" -> in the GAN
7. p6 "piano playing audio on our own..." -> piano playing on its own (or even just rephrase the sentence - collect 4.5 hours of audio of solo piano)
8. p7 "We apply source separation to the audios divide them into ..." -> we apply source separation to the audio data then divide each track into 20 second...
9. p7 If your piano transcription model was worse than Hawthorne, why didn't you use it? It would have been fine to say you can't reproduce their model if it is not available, but instead you say that 'according to out observation [it] is strong enough' which comes across quite weakly.
10. p8 "in a quiet environment with proper microphone volume" -> headphone volume?
11. p8 "improv" - I think this sentence trailed off prematurely!



**Experience Assessment:**

I have read many papers in this area.

**Review Assessment: Checking Correctness Of Derivations And Theory:**

N/A

**Review Assessment: Checking Correctness Of Experiments:**

I assessed the sensibility of the experiments.

**Review Assessment: Thoroughness In Paper Reading:**

I made a quick assessment of this paper.

---

> ### Author Response · Authors · 2019-11-13
> **Response to the Review #6**
>
> Thank you for the valuable comments. We address the issues and questions raised by the reviewer in the following comments.
>
>
> $\textbf{1.}$ The motivation is slightly lacking ..., and there is a lack of discussion about which setting makes for better singing voice generation. Also, there is no comparison with other methods ...
>
> $\textbf{Ans:}$
> We have revised the paper to address these issues:
> 1. Including clear motivations in Introduction
> 2. Including in Appendix some experimental results of using different GAN losses.
> 3. Expanding Section 4 by adding two baseline synthesis methods with the approach you suggested.
>
>
> $\textbf{2.}$ Regarding using MIR-1k
>
> $\textbf{Ans:}$
> MIR-1k indeed contains cleaner vocal-accompaniment pairs. However, only a few of the accompaniment tracks in MIR-1k contain piano playing. In the current setting of accompanied singer, the accompaniment has to contain piano in order to use them as the condition. Therefore, MIR-1k is not able to be used as the training data currently.
>
>
> $\textbf{3.}$ ... training data ... is everything Jazz? Are the accompaniments exclusively piano ...? Is there any difference between the training and test domain.
>
> $\textbf{Ans:}$
>
> Singer models - Training data:
> Jazz music tracks from Youtube containing both singing and piano
> Everything is Jazz. Usually the audios in this set also contain instruments other than piano, such as drums, bass, and guitar.
>
> Singer models - Testing data:
> Jazz music tracks from Jamendo. We collect those Jazz music tracks containing piano, but some of them also contain other instruments and vocals. Therefore, we still have to apply source separation to them to separate the piano tracks and transcribe them. The training and test data are both Jazz with piano.
>
> Source separation model:
> MUSDB (four sources: bass, drums, vocals, other) + 4.5 hours of solo piano from Youtube as the fifth  “piano” source. Furthermore, the “other” tracks in MUSDB that contain piano playing are removed to avoid confusion between “other” and “piano.”
>
> To make it clearer, we have added Table 5 in appendix to list the datasets and their usage.
>
>
> $\textbf{4.}$ Regarding the split of the validation set
>
> $\textbf{Ans:}$
> The validation split contains clips randomly sampled from all the clips in the training set, so the same track can be in both the training and validation splits. We did this for the following reason. The female jazz tracks collected from Youtube are all long, ranging from 35 minutes to two hours. If we split training and validation without track overlapping, the validation split would either include a large portion of the training set or contain clips only from very few (1 or 2) tracks. Therefore, we decide to split them with track overlapping.
>
>
> $\textbf{5.}$ Regarding the MelNet
>
> $\textbf{Ans:}$
> After reading the paper, we found that MelNet is indeed related to our work.
>
> First, MelNet and this paper both work on time-frequency representations, so the techniques to capture structural information in the TF representations might be used in our models too. We have referred to it in the conclusion as a direction of the future work.
>
> Second, MelNet and our models both accommodate the unconditional and conditional generation. The unconditional generated speeches of MelNet are especially impressive to us as the phonemes are intelligible. It shows that MelNet can capture latent properties of speech. It is a promising way for us to explore.
>
> Third, there is also a major difference between MelNet and this paper. MelNet is applied to piano and speech, while we explore the generation of singing voices. Singing voice generation is somewhere between speech generation and music generation, so whether the techniques in MelNet can apply to our tasks still require investigations.
>
>
> $\textbf{6.}$ My main criticism is in relation to the evaluation.
>
> $\textbf{Ans:}$
> To address this issue, we have added two baseline synthesis methods (Sinsy and Synthesizer V) in a new MOS study. In addition, we have also expanded Table 1 to include the objective metrics of the training data.
>
>
> $\textbf{7.}$
> For Table 1,  .... explains the differences in scores for the different model settings
>
> $\textbf{Ans:}$
> We have included the objective metrics computed from the training data and the Sinsy singing as well as the discussion about them.
>
> Furthermore, we have updated the method of computing vocalness so that now the vocalness takes into account both the vocal activation and the pitch range of singing voices. The details are in Section 4.3.
>
>
> $\textbf{8.}$ Tables 1 and 2, and the provided audio samples have no context, so I cant make a conclusion. If this issue and motivation was addressed I would likely vote to accept the paper.
>
> $\textbf{Ans:}$
> We have updated Table 1 and added Table 3 so that our methods can be compared with other methods.
> Audio samples of other synthesis methods are also included in the paper: https://bit.ly/2qNrekv

---

### Author Response · Authors · 2019-11-14
**Summary of the paper update**

Thank you very much for all of your valuable comments.  We have taken your constructive suggestions seriously and revise part of the paper accordingly.  Below is a summary of the major changes:

* Two new baseline methods by using well-known singing voice synthesis systems, Sinsy and Synthesizer V (Described in Section 4.2)

* A new user study that compares our model with the two new baseline methods (Table 3 and Section 4.4)

* Expanding Table 1 and Section 4.3 to include the objective metrics of Sinsy baseline as well as the objective metrics computed from the training data

* Expanding the Introduction to further discuss the motivations and possible usages of our methods

---

### Decision · Program_Chairs · 2019-12-19

**Decision:**

Reject

**Comment:**

Main content:

Blind review #1 summarizes it well:

his paper claims to be the first to tackle unconditional singing voice generation. It is noted that previous singing voice generation approaches leverage explicit pitch information (either of an accompaniment via a score or for the voice itself), and/or specified lyrics the voice should sing. The authors first create their own dataset of singing voice data with accompaniments, then use a GAN to generate singing voice waveforms in three different settings:
1) Free singer - only noise as input, completely unconditional singing sampling
2) Accompanied singer - Providing the accompaniment *waveform* (not symbolic data like a score - the model needs to learn how to transcribe to use this information) as a condition for the singing voice
3) Solo singer - The same setting as 1 but the model first generates an accompaniment then, from that, generates singing voice

--

Discussion:

The reviews generally point out that while a lot of new work has been done, this paper bites off too much at once: it tackles many different open problems, in a generative art domain where evaluation is subjective.

--

Recommendation and justification:

This paper is a weak reject, not because it is uninteresting or bad work, but because the ambitious scope is really too large for a single conference paper. In a more specialized conference like ISMIR, it would still have a good chance. The authors should break it down into conference sized chunks, and address more of the reviewer comments in each chunk.